# Standardizing workflows in imaging transcriptomics with the abagen toolbox

**Ross D Markello[1]\*, Aurina Arnatkeviciute[2], Jean-Baptiste Poline[1], Ben D Fulcher[3], Alex Fornito[2], Bratislav Misic[1]\***

[1]McConnell Brain Imaging Centre, Montreal Neurological Institute, McGill University, Montreal, Canada; [2]School of Psychological Sciences & Monash Biomedical Imaging, Monash University, Clayton, Australia; [3]School of Physics, University of Sydney, Sydney, Australia

**Abstract** Gene expression fundamentally shapes the structural and functional architecture of the human brain. Open-access transcriptomic datasets like the Allen Human Brain Atlas provide an unprecedented ability to examine these mechanisms in vivo; however, a lack of standardization across research groups has given rise to myriad processing pipelines for using these data. Here, we develop the abagen toolbox, an open-access software package for working with transcriptomic data, and use it to examine how methodological variability influences the outcomes of research using the Allen Human Brain Atlas. Applying three prototypical analyses to the outputs of 750,000 unique processing pipelines, we find that choice of pipeline has a large impact on research findings, with parameters commonly varied in the literature influencing correlations between derived gene expression and other imaging phenotypes by as much as $\rho \geq 1.0$. Our results further reveal an ordering of parameter importance, with processing steps that influence gene normalization yielding the greatest impact on downstream statistical inferences and conclusions. The presented work and the development of the abagen toolbox lay the foundation for more standardized and systematic research in imaging transcriptomics, and will help to advance future understanding of the influence of gene expression in the human brain.

**\*For correspondence:**
ross.markello@mail.mcgill.ca (RDM);
bratislav.misic@mcgill.ca (BM)

## Editor's evaluation

This paper will be of interest to scientists studying the large-scale transcriptomic organization of the human brain, and in particular those who have used or plan to use the Allen Human Brain Atlas dataset. The study is well-motivated and novel. The most striking finding is the magnitude of variability that is introduced by different data processing decisions. The open-source software described in this study is comprehensive, well documented, and is an important contribution to the field.

## Introduction

Technologies like magnetic resonance imaging (MRI) provide unique insights into macroscopic brain structure and function in vivo. Modern research increasingly emphasizes how microscale attributes, such as gene expression, influence these imaging-derived phenotypes (*Fornito et al., 2019*; *Arnatkeviciute et al., 2019*; *Arnatkevičiūtė et al., 2021*). Gene expression is particularly useful as it is a fundamental molecular phenotype that can be plausibly linked to the function of biological pathways (*Whitaker et al., 2016*; *Seidlitz et al., 2018*), protein synthesis (*Zheng et al., 2019*), receptor distributions (*Beliveau et al., 2017*; *Nørgaard et al., 2021*; *Shine et al., 2019*; *Deco et al., 2020*; *Preller et al., 2018*), and cell types (*Hansen et al., 2021*; *Anderson et al., 2020b*; *Anderson et al., 2018*; *Seidlitz et al., 2020*; *Gao et al., 2020*). However, researchers looking to bridge these macro- and

microscopic phenotypes must overcome multiple challenges. Although there are numerous technical and analytic considerations, one foundational issue is that acquiring high-quality transcriptomic data from the human brain is both costly and highly invasive, requiring budgets far greater than most typical neuroimaging studies and restrictive access to tissue from post-mortem donors or cranial surgical patients. As such, researchers must often rely on freely available repositories of gene expression data.

There exist multiple open-access repositories for gene expression in the human brain, including BrainSpan (*Miller et al., 2014*; *Kang et al., 2011*) and PsychENCODE (*Gandal et al., 2018*; *Li et al., 2018*; *Wang et al., 2018*; among others: *Sousa et al., 2017*; *Darmanis et al., 2015*; *Lake et al., 2016*); however, these datasets generally provide relatively sparse anatomical coverage, limiting the types of analyses that can be performed. Thus, researchers who aim to compare transcriptomic expression with whole-brain imaging-derived phenotypes have primarily relied on the Allen Human Brain Atlas (AHBA; *Hawrylycz et al., 2012*; *Hawrylycz et al., 2015*). Initially released in 2010, the AHBA remains the most spatially comprehensive dataset of its kind. Derived from bulk microarray analysis of tissue samples obtained from six donors, the AHBA provides expression data for more than 20,000 genes across 3702 brain areas in MRI-derived stereotactic space. With its superior resolution, the AHBA has significantly contributed to the emergence of the field of imaging transcriptomics (*Fornito et al., 2019*), enabling dozens of studies over the past decade examining relationships between gene expression and an array of macroscale imaging attributes, including cortical thickness (*Shin et al., 2018*), myelination (*Burt et al., 2018*), developmental brain maturation (*Whitaker et al., 2016*; *Kirsch and Chechik, 2016*), structural brain networks (*Seidlitz et al., 2018*; *Romero-Garcia et al., 2018*; *Arnatkevičiūtė et al., 2020*), functional brain networks (*Richiardi et al., 2015*; *Krienen et al., 2016*; *Vértes et al., 2016*), and human cognition (*Fox et al., 2014*; *Hansen et al., 2021*). The AHBA has also highlighted the importance of whole-brain gene expression in neurological and psychiatric diseases, where it has become increasingly clear that transcriptional pathways play a critical role in shaping the broader dynamics of disease progression and emergent symptomatology (*Zheng et al., 2019*; *Shafiei et al., 2021*; *Henderson et al., 2019*; *Vogel et al., 2020*; *Rittman et al., 2016*; *Anderson et al., 2020a*; *Romme et al., 2017*; *McColgan et al., 2018*; *Morgan et al., 2019*).

Since its release, several software toolboxes have been developed to help researchers use transcriptional data from the AHBA (*French and Paus, 2015*; *Gorgolewski et al., 2015*; *Rittman et al., 2017*; *Rizzo et al., 2016*); however, these tools often focus primarily on facilitating integration of the AHBA with neuroimaging data, offering limited if any functionality for modifying how the data are processed prior to analysis. Instead, a recent comprehensive review revealed that many research groups have opted to develop their own processing pipelines for the AHBA (*Arnatkeviciute et al., 2019*). Unfortunately, as there are no field-accepted standards for processing imaging transcriptomic data, the generated pipelines vary substantially across groups.

The extent to which such processing variability affects analytic outcomes from the AHBA remains unknown. Indeed, over the past decade neuroimaging research has shown that methodological variability can have broad influences on analyses using structural MRI (*Bhagwat et al., 2021*; *Kharabian Masouleh et al., 2020*), diffusion MRI (*Oldham et al., 2020*; *Maier-Hein et al., 2017*; *Schilling et al., 2019*), task fMRI (*Carp, 2012*; *Botvinik-Nezer et al., 2020*), and resting-state fMRI (*Parkes et al., 2018*; *Ciric et al., 2017*). Although researchers are beginning to grapple with the consequences of this variability, the lack of baseline gene expression datasets against which to compare new results impedes the development of standardized practices. In these situations, some researchers have proposed performing 'multiverse' analyses (*Steegen et al., 2016*; *Dragicevic et al., 2019*), wherein all possible permutations of data processing are analyzed and the full range of analytic results reported. Although such analyses can be computationally intensive, they offer a path to understand how processing choices impact statistical inferences and conclusions, and provide a mechanism by which to help researchers converge on an optimal pipeline.

Here, we comprehensively investigate how different processing choices influence the results of analyses using the AHBA. First, we develop an open-source Python toolbox, abagen, that collates all possible processing parameters into a set of turn-key workflows, optimized for flexibility and ease-of-use. We then use the toolbox to process the AHBA through approximately 750,000 unique pipelines. Across three prototypical imaging transcriptomic analyses, we examine whether and how these different processing options modify derived statistical estimates and quantify the relative importance of each option. Next, we replicate a curated set of processing pipelines from the literature to assess

how previously reported findings compare to the full range of potential outcomes observed across all examined pipelines. Finally, we end with a set of recommendations, integrated directly into the developed abagen toolbox, to promote standardized use of the AHBA in future work.

## Results

We introduce the abagen toolbox, an open-access software package designed to streamline processing and preparation of the AHBA for integration with neuroimaging data (*Markello et al., 2021c*, available at https://github.com/rmarkello/abagen; *Markello, 2021b* copy archived at swh:1:rev:2aeab5bd0f-147fa76b488645e148a1c18095378d). Supporting several workflows, abagen offers functionality for

**Table 1.** Abagen pipeline options.
Overview of 17 options to be considered when processing the AHBA data. The *Choices* column indicates the number of parameters explored in the current report (numerator) and the total number of parameters possible for the given option (denominator). A denominator of *n* indicates a hypothetically near-infinite parameter space. The *Description* column gives a brief overview of the processing choice; for more detail refer to the relevant section in Materials and methods*: Gene expression pipelines*.

| Option | Choices | Description |
|---|---|---|
| Volumetric or surface atlas | 2/2 | Whether to use a volumetric or surface representation of the atlas |
| Individualized or group atlas | 1/2 | Whether to use individualized donor-specific atlases or a group-level atlas |
| Use non-linear MNI coordinates | 2/2 | Whether to use updated MNI coordinates provided by alleninf package |
| Mirror samples across L/R hemisphere | 3/4 | Whether to mirror (i.e., duplicate) samples across hemisphere boundary |
| Update probe-to-gene annotations | 2/2 | Whether to update probe annotations |
| Intensity-based filtering threshold | 3/$n$ | Threshold for intensity-based filtering of probes |
| Inter-areal similarity threshold | 1/$n$ | Threshold for removing samples with low inter-areal correspondence |
| Probe selection method | 6/8 | Method by which to select which probe(s) should represent a given gene |
| Donor-specific probe selection | 3/3 | How specified probe selection should integrate data from different donors |
| Missing data method | 2/3 | How to handle when brain regions are not assigned expression data |
| Sample-to-region matching tolerance | 3/$n$ | Distance tolerance for matching tissue samples to atlas brain regions |
| Sample normalization method | 3/10 | Method for normalizing tissue samples (across genes) |
| Gene normalization method | 3/10 | Method for normalizing genes (across tissue samples) |
| Normalize only matched samples | 2/2 | Whether to perform gene normalization for all versus matched samples |
| Normalizing discrete structures | 2/2 | Whether to perform gene normalization within structural classes |
| Sample-to-region combination method | 2/2 | Whether to aggregate tissue samples in regions within or across donors |
| Sample-to-region combination metric | 2/2 | Metric for aggregating tissue samples into atlas brain regions |

an array of analyses and has already been used in several peer-reviewed publications and preprints (*Shafiei et al., 2020*; *Hansen et al., 2021*; *Shafiei et al., 2021*; *Brown et al., 2021*; *Park et al., 2021*; *Valk et al., 2021*; *Zhao et al., 2020*; *Benkarim et al., 2020*; *Ding et al., 2021*; *Park et al., 2020*; *Lariviere et al., 2020*; *Martins et al., 2021*). The primary workflow, used to generate regional gene expression matrices, integrates 17 distinct processing steps that have previously been employed by research groups throughout the published literature (*Table 1*). We refer to each unique set of processing choices and parameters as a 'pipeline'. The following results use abagen to investigate how variable application of these processing steps can impact analyses of AHBA data.

## Processing choices influence transcriptomic analyses

To understand how choices made during the processing of AHBA data impact downstream analyses, we enumerated 17 decision points (i.e. processing steps or options) that have been modified and used in the literature (*Table 1*). From these 17 steps we implemented 746,496 distinct processing pipelines, where each pipeline parcellated microarray expression from the AHBA with the Desikan-Killiany atlas (*Desikan et al., 2006*) to generate a unique brain region-by-gene expression matrix.

Analyses of expression data from the AHBA can be grouped into one of three broad classes (*Fornito et al., 2019*): correlated gene expression analyses, gene co-expression analyses, and regional gene expression analyses. Correlated gene expression analyses examine the correlation between brain regions across genes, yielding a symmetric region × region matrix (similar to a functional connectivity matrix). Gene co-expression analyses, on the other hand, examine the correlation between genes across brain regions, yielding a symmetric gene × gene matrix. Finally, regional gene expression analyses examine the expression patterns of specific genes or gene sets in relation to other imaging-derived phenotypes.

To examine how differences in processing choices may impact both the expression matrices generated from the different pipelines and derived statistical estimates we ran one analysis from each of these classes on the matrices generated by each processing pipeline. Notably, these analyses are either direct reproductions or variations of analyses that have been previously published (*Arnatkeviciute et al., 2019*; *Oldham et al., 2008*; *Hawrylycz et al., 2012*; *Burt et al., 2018*). Although there is no ground truth for any of these analyses, findings from previous work offer some context for interpreting the observed results (i.e. data from other species and other modalities; *Lau et al., 2021*). Nonetheless, we primarily focus on highlighting the potential variability resulting from different processing pipelines.

### Correlated gene expression (CGE)

First, we separately correlated the rows of each expression matrix to generate symmetric region × region 'correlated gene expression' matrices, indicating the similarity of gene expression profiles between different brain regions (*Figure 1a*). Previous work in other species has reliably observed that transcriptional similarity in the brain decays with increasing separation distance (*Fulcher et al., 2019*; *Lau et al., 2021*). This distance-dependent relationship is an expected feature due to the functional specialization of brain regions, and is consistent with other imaging-derived phenotypes in humans (*Roberts et al., 2016*; *Goulas et al., 2019*; *Betzel and Bassett, 2018*; *Mišić et al., 2014*; *Shafiei et al., 2020*; *Horvát et al., 2016*). We assessed this relationship by extracting the upper triangle of the correlated gene expression matrices and correlating them with the upper triangle of a regional distance matrix, derived by computing the average Euclidean distance between brain region centroids in the Desikan-Killiany atlas (*Figure 1a*, left panel). Although previous work has highlighted that this relationship is exponential (*Arnatkeviciute et al., 2019*), we computed the Spearman correlation as both statistics should exhibit similar variability across pipelines and the latter is less computationally expensive.

### Gene co-expression (GCE)

For the second type of analysis we separately correlated the columns of each expression matrix to generate gene × gene 'co-expression' (GCE) matrices, indicating the similarity in spatial expression patterns between all pairs of genes (*Figure 1a*). A significant body of research has shown that genes tend to form functional communities, exhibiting synchronized expression patterns across space and time (*Oldham et al., 2008*), such that gene co-expression patterns tend to be more similar within than

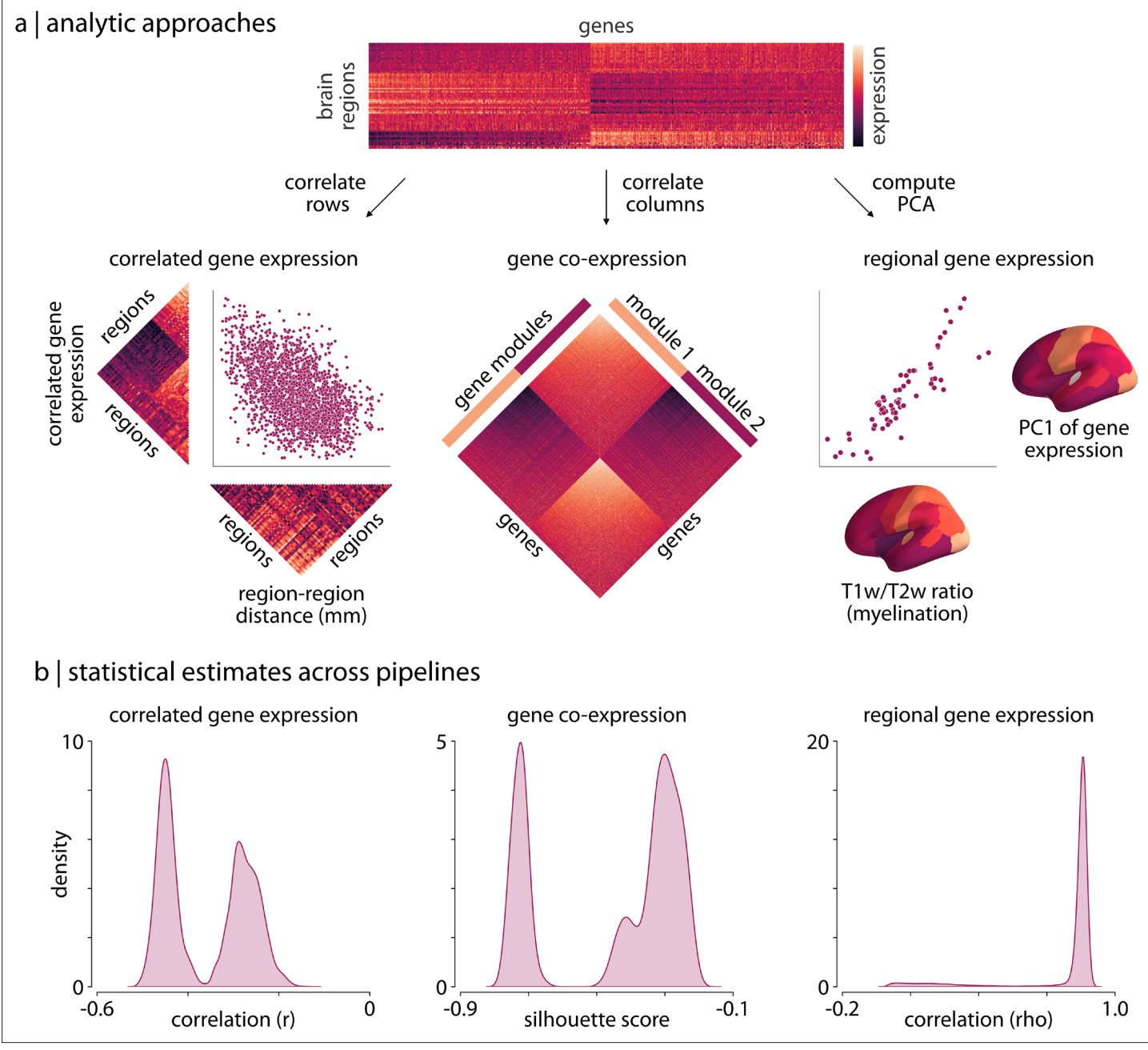

**Figure 1.** Processing choices influence transcriptomic analyses. (**a**) Examples of the three analyses used to assess differences in gene expression matrices generated by transcriptomic pipelines. First row: a depiction of the region-by-gene expression matrix generated from one of the 746,496 tested processing pipelines. Second row, left: we compute the correlation between rows of each matrix to generate a symmetric region × region CGE matrix. We then compute the correlation between the upper triangle of this CGE matrix and the upper triangle of a regional distance matrix to examine the degree to which CGE decays with increasing distance between regions (**Arnatkeviciute et al., 2019**). Second row, middle: we compute the Euclidean distance between columns of each matrix to generate a gene × gene GCE matrix. We use previously defined functional gene communities (**Oldham et al., 2008**) to compute a silhouette score for this GCE matrix to investigate whether genes within a module have more similar patterns of spatial expression than genes between modules. Second row, right: the first principal component is extracted from the RGE matrix. We compute the correlation between this principal component and the whole-brain T1w/T2w ratio (**Burt et al., 2018**) to understand how closely these maps covary across the brain. (**b**) The full statistical distributions from each of the three analyses for all 746,496 pipelines. Left panel: Spearman correlation values, $\rho$, from the CGE analyses. Middle panel: silhouette scores from the GCE analyses. Right panel: Spearman correlation coefficients, $\rho$, from the RGE analyses. CGE: correlated gene expression; GCE: gene co-expression; RGE: regional gene expression.

between such communities. Here, we obtained a set of gene community assignments derived for the brain from a previously studied human transcriptomic dataset (*Oldham et al., 2008*). We used these community assignments to calculate a silhouette score (*Rousseeuw, 1987*) for the gene co-expression matrices generated by each pipeline, measuring how well these communities represented the derived co-expression patterns (*Figure 1a*, middle panel).

### Regional gene expression (RGE)

For the third type of transcriptomic analysis, we focused on regional correlations between gene expression measures and an MRI-derived phenotype. Our regional expression measure was defined by computing the first principal component of the region-by-gene expression matrix, representing the axis of maximum spatial variation of gene expression in the brain observed under a given AHBA processing pipeline. As gene expression fundamentally shapes the structure and function of the human brain, it is likely that this principal component may exhibit similar spatial variability to other imaging-derived measures. Recent work has highlighted that the T1w/T2w ratio is a robust phenotype that exhibits patterns of regional variation consistent with other microstructural and functional properties (*Gao et al., 2020*; *Burt et al., 2018*; *Demirtaş et al., 2019*; *Fulcher et al., 2019*). We therefore correlated the first principal component of gene expression with the whole-brain T1w/T2w ratio (*Figure 1a*, right panel), measuring the extent to which these values covary across the cortex.

### Pipeline distributions

Results from these three analyses reveal that choice of processing pipeline dramatically influences derived statistical estimates (i.e. the CGE-distance correlation, the gene co-expression silhouette score, and the spatial correlations between gene PC1 and whole-brain T1w/T2w ratio; *Figure 1b*). We observe that all three of the generated distributions of statistical estimates across the 746,496 pipelines have wide ranges (correlated gene expression: [-0.51,–0.13]; gene co-expression: [-0.78,–0.18]; regional gene expression: [0.00, 0.90]) and are either bimodal (*Figure 1b*, left/middle panels) or heavily skewed (*Figure 1b*, right panel).

Since there is no ground truth for these analyses we cannot quantitatively assess whether some pipelines are more or less accurate than others. However, there is strong qualitative evidence to suggest that correlated gene expression should be lower between brain regions that are farther apart (*Arnatkeviciute et al., 2019*; *Krienen et al., 2016*; *Richiardi et al., 2015*; *Fulcher et al., 2019*; *Lau et al., 2021*). It is notable, then, that the distribution of distance-dependent estimates is so strongly bimodal (splitting at $r \approx -0.4$), suggesting two very different perspectives on the size of this effect (*Figure 1a and b*, left panels). As increasingly-detailed single-cell transcriptional data become available (e.g. *Yao et al., 2021*) we may be able to use these estimates to determine accuracy; for now, we simply note that even for this estimate with strong biological priors we see considerable variability.

Similar variability can be observed for the other two analyses. While all the pipelines demonstrate relatively poor fit of gene communities to the derived gene co-expression matrices (refer to Materials and methods: *Analytic approaches* for information on why this is not unexpected), we observe that a portion of the pipelines yield far worse correspondence (*Figure 1a and b*, middle panels). Moreover, while the correlations between gene PC1 and whole-brain T1w/T2w ratio are largely consistent across pipelines, there are a small group of pipelines that yield correlations that deviate by $\rho \approx 1.0$. Notably, the parameter choices for these pipelines are not pathological—that is, their use could be justified—and, as we discuss later (see *Results: Variability in parameter importance*), modifying just one parameter setting can yield changes in effect sizes within this range.

Collectively, we find that for all three of these analyses there is substantial variability in the statistical estimates generated by different processing pipelines, and this variability is large enough that, across pipelines, it has a meaningful difference in the potential inferences and conclusions that can be drawn.

## Variability in parameter importance

Next, we quantified the relative importance of different processing steps and parameters on our three derived statistical estimates. While researchers must ultimately make choices for each of the steps individually when processing AHBA data, we wanted to investigate whether unique choices have

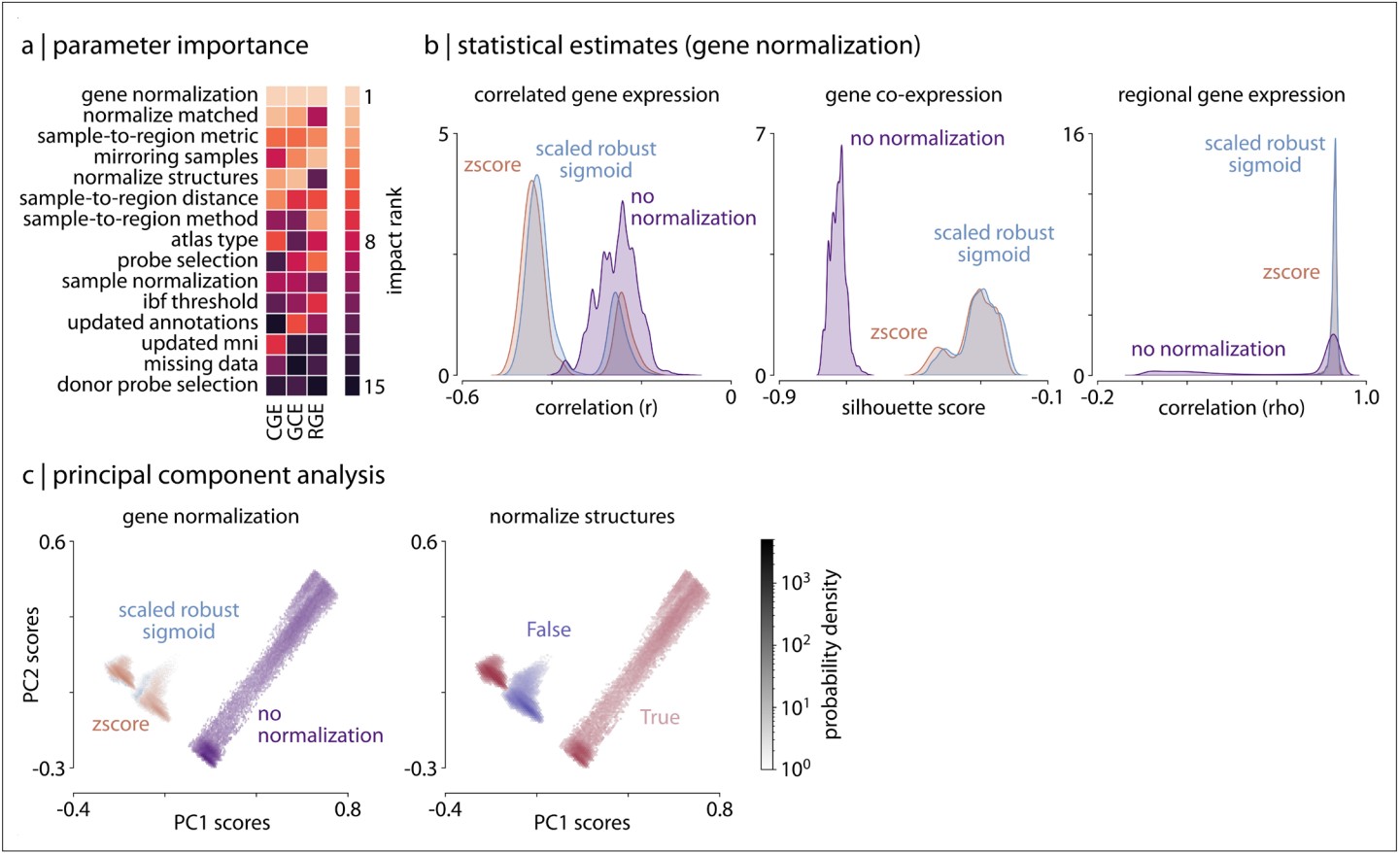

**Figure 2.** Parameter choice differentially impacts statistical estimates. (**a**) Rank of the relative importance for each parameter (*y*-axis) across all three analyses (*x*-axis). Warmer colors indicate parameters that have a greater influence on statistical estimates. (**b**) Statistical distributions from the three analyses, shown as kernel density plots, separated by choice of gene normalization method (the most impactful parameter as shown in panel a). (**c**) Density plots of the statistical estimates for all 746,496 pipelines shown along the first two principal components, derived from the 746,496 (pipeline) x 3 (statistical estimates) matrix, representing how different the statistical estimates from each of the three analyses are relative to other pipelines. Left panel: pipelines are colored based on choice of gene normalization method, where each color represents 1/3 of the pipelines. Here, the pipelines in which no normalization was applied (purple) are distinguished from those in which some form of normalization was applied (blue and brown). Right panel: pipelines are colored based on whether gene normalization was performed within (True, red) or across (False, purple) structural classes (i.e. cortex, subcortex/brainstem, cerebellum; see *Materials and methods: Gene expression pipelines* for more information).

distinct influences. Moreover, which parameters are most important may differ based on the type of analysis performed.

We investigated parameter importance by calculating a distribution of difference scores for each parameter, measuring the extent to which changing each parameter—holding all other parameters constant—influences the derived statistical metrics from each of the three analyses. For example, given a processing parameter with two choices this procedure yielded a distribution of $N/2$ difference scores per analysis, where $N$ is the total number of pipelines (i.e. $746,496/2 = 373,248$). We averaged these distributions separately for each analysis to generate a single, summary 'impact score' for each processing step, which we then rank-ordered independently for each analysis.

We find considerable agreement in which parameters are the most impactful across analyses (*Figure 2a*): the most influential processing steps often involve procedures that influence the gene normalization process in some way (e.g. gene normalization method, normalizing only matched samples; *Figure 2b*). On the other hand, among the least impactful parameters are choices concerning donor-specific probe selection and handling of missing data. It is worth noting that of the probe selection methods tested in the current manuscript (i.e. max intensity, correlation intensity, correlation variance, differential stability, RNAseq correlation, and averaging), three of the six all render the choice of donor-specific probe selection redundant. In other words, these three methods are mutually exclusive with choice of donor-specific probe selection, potentially confounding our ability to measure the real

influence of this parameter. We also highlight that choice of atlas may influence the impact of missing data handling: since the Desikan-Killiany atlas is a relatively low-resolution atlas (68 nodes), expression matrices generated from the tested pipelines are missing, at most, data for two brain regions. It is possible that handling of missing data may be more important when higher-resolution parcellations are employed. That is, while some parameters do not appear to affect our results *in aggregate*, there are potentially specific research questions where these parameters could play an important and impactful role.

To investigate those parameters that did play an influential role in the current analyses, we visualized their impact by examining the statistical distributions from each analysis separated by the different parameter choices (shown in *Figure 2b* for gene normalization method). Dividing the distributions in this way highlights how strongly parameter choice can influence the outcomes of the analyses: for example, when no gene normalization is employed the resulting estimates are dramatically shifted from those generated by pipelines that employed some form of normalization (*Figure 2b*; no normalization: purple distribution). Indeed, the bimodality and skew observed in the full statistical distributions for the analyses (*Figure 1b*) is almost entirely explained by this single parameter choice.

To investigate more qualitative differences in how parameter choice influences the processing pipelines we performed a principal component analysis (PCA) on the matrix of statistical estimates from the three analyses (i.e. the $746,496 \times 3$ pipeline-by-analysis matrix). We extracted the first two principal components from the statistical estimate matrix (variance explained: PC1 = 70%, PC2 = 26%) and examined how pipeline scores were distributed along these axes (*Figure 2c*). Delineating the distribution of pipelines based on parameter choice underscores how these options impact the separability of resulting statistical estimates. Reinforcing results presented above, we find that the choice of gene normalization method distinguishes the one-third of pipelines with no normalization (purple) from the remaining two-thirds that applied some form of normalization (blue and brown; *Figure 2c*, left). It is clear from the distribution of pipelines, however, that other processing choices interact with this parameter. For example, plotting the pipelines by whether the gene normalization was performed separately on samples within each structural class (i.e. cerebral cortex, subcortex, cerebellum) rather than across all tissue samples further delineates the pipelines that applied gene normalization into two distinct clusters (*Figure 2c*, right).

These results reveal how different processing steps are grouped in terms of their importance to analyses of the AHBA, with some groups demonstrating greater potential impact. Broadly, parameters modifying normalization are the most important, followed by parameters influencing how tissue samples are matched to brain regions, and finally parameters impacting probe selection. Moreover, we find that choices within each processing step do not all have an equivalent impact on derived estimates (i.e. performing no gene normalization has a much greater influence than choosing between the two other forms of normalization tested).

## Reproducing published analyses

The previous subsections demonstrate variability across the complete range of reasonable processing pipelines; however, many of these pipelines have not yet been used in practice. To investigate whether the subset of pipelines that have already been implemented in the published literature display similar variability, we used abagen to reproduce the processing procedures from nine peer-reviewed articles that (1) are highly-cited within the field, (2) highlight a wide range of processing options, and (3) sufficiently describe their processing pipelines such that they could be reproduced. We explored how different the gene expression values and statistical outcomes generated by these published pipelines were (*Hawrylycz et al., 2015*; *French and Paus, 2015*; *Whitaker et al., 2016*; *Krienen et al., 2016*; *Anderson et al., 2018*; *Burt et al., 2018*; *Romero-Garcia et al., 2018*; *Anderson et al., 2020b*; *Liu et al., 2020*). To ensure comparability, we standardized the choice of brain parcellation across pipelines, using the Desikan-Killiany atlas in all instances. The pipelines were used to generate nine region-by-gene expression matrices, which were then subjected to the same three analyses described previously.

In reproducing the pipelines we note important differences in processing parameter selection (*Figure 3a*), and find that this variability results in slight discrepancies between gene expression values generated by the pipelines. For example, looking at the distribution of cortical somatostatin (SST), a gene discussed heavily in *Anderson et al., 2020b* where it used as a proxy for somatostatin

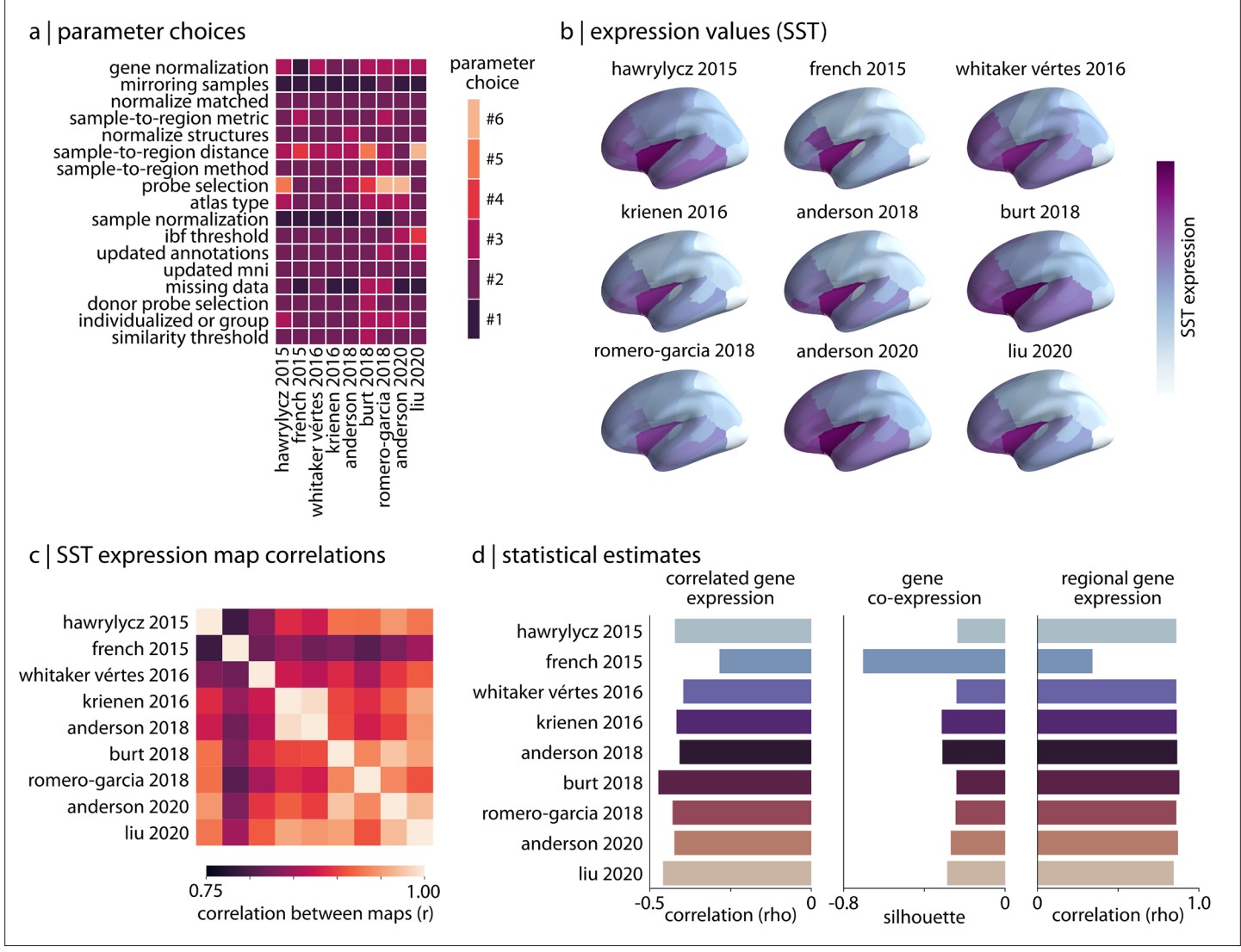

**Figure 3.** Reproducing published pipelines. (**a**) Parameter choices used in the reproduction of published pipelines. Processing steps with categorical choices (e.g., gene normalization) were converted to numerical choices for display purposes only. These choices reflect the range of choices enumerated in Table 1. (**b**) Relative expression values of cortical somatostatin (SST) generated by each of the reproduced pipelines. Value ranges vary based on pipeline processing options. (**c**) The Pearson correlation between the cortical somatostatin (SST) maps generated by the nine pipelines shown in panel (**b**). (**d**) Statistical estimates from the three analyses described in *Materials and methods: Analytic approaches* applied to expression data from each of the published pipelines.

interneuron density (*Fulcher, 2019*), we observe some variation between pipelines (*Figure 3b and c*). Although we find moderate consistency in the statistical estimates generated by the pipelines, there are important differences (ranges: correlated gene expression [-0.49,–0.28], gene co-expression [-0.70,–0.24], regional gene expression [0.34, 0.88]; *Figure 3c*). One outlier is the single pipeline that did not appear to implement any form of gene normalization (*French and Paus, 2015*), supporting earlier results demonstrating the importance of this processing step on downstream expression estimates. This is potentially notable as the processed expression data from this pipeline were made openly available and have been used in analyses by other researchers (e.g. *Sepulcre et al., 2018*; *Beliveau et al., 2017*).

Given that imaging transcriptomics is still relatively new and there has been limited work addressing best practices in the field (*Arnatkeviciute et al., 2019*), these results stress the importance of standardization in use of the AHBA among research groups. Although variation in processing can ostensibly lead to similar inferences in specific analyses, even minor differences in processing choices

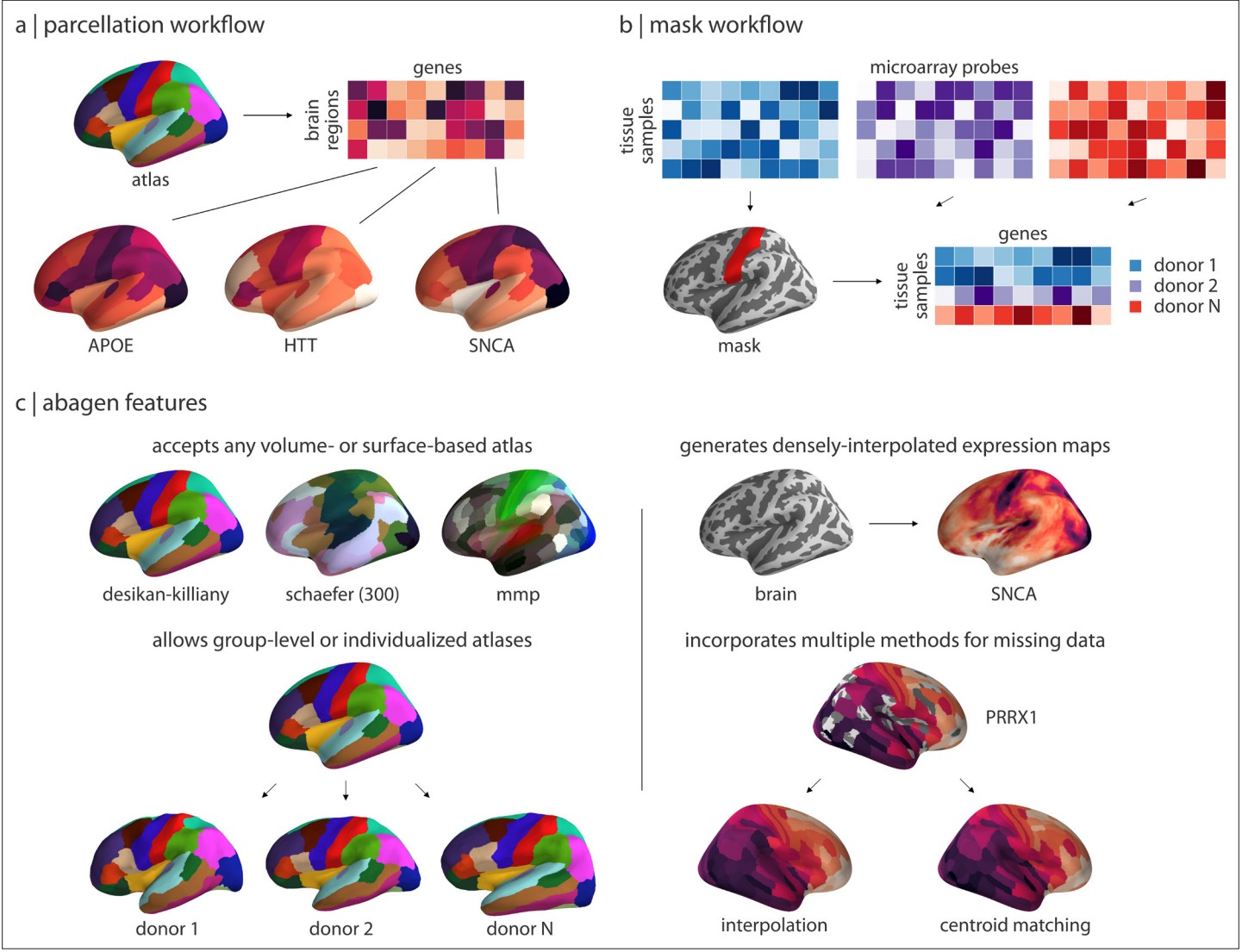

**Figure 4.** Workflows and features in the abagen toolbox. (**a**) The primary workflow of abagen, used in the reported analyses, accepts a brain atlas and returns a parcellated brain-region-by-gene expression matrix. (**b**) An alternative abagen workflow accepts a regional mask and returns a processed tissue-sample-by-gene expression matrix, for all tissue samples from the six AHBA donors that fall within boundaries of the mask. (**c**) Examples of selected features from the abagen workflows and additional toolbox functionality. Top left: examples of some commonly-used atlases that can be employed with the parcellation workflow shown in panel (**a**). Bottom left: abagen can accept either standard atlases (i.e. in MNI space) or atlases defined in the space of the six individual donors from the AHBA. Top right: an additional workflow available in abagen can be used to generate densely-interpolated expression maps from AHBA data using a k-nearest neighbors interpolation algorithm. Bottom right: using high-resolution atlases in the parcellation workflow (panel a) may result in some parcels being assigned no expression data; abagen supports two methods for assigning values to such regions.

consistently yield measurable discrepancies in derived expression data. Without proper standardization, these differences will compound and become more problematic as the field continues to grow.

## Standardized processing and reporting with the abagen toolbox

Across all of our analyses we find that choice of processing steps and parameters can have a strong influence on the statistical outcomes of research with the AHBA. Here, we briefly highlight features that we have integrated into the abagen toolbox to facilitate standardization in future research.

The abagen toolbox supports two use-case driven workflows: (1) a workflow that accepts an atlas and returns a parcellated, preprocessed regional gene expression matrix (*Figure 4a*); and, (2) a workflow that accepts a mask and returns preprocessed expression data for all tissue samples within the

mask (*Figure 4b*). Workflows can be called via a single line of code from either the command line or Python terminal, and take approximately one minute to run with default settings using the Desikan-Killiany atlas. The main output of abagen is a single brain region (or tissue sample) × gene expression matrix. Changing the parameters may modify the shape of the matrix (e.g. different atlases will yield different numbers of regions or samples) or different values (e.g. different processing choices may yield different numbers of genes), but not the structure. The outputs of these workflows can be used generally to examine the three prototypical research questions enabled by the AHBA: correlated gene expression, gene co-expression, and regional expression of genes of interest more broadly (*Fornito et al., 2019*). Beyond its primary workflows, abagen has additional functionality for post-processing the AHBA data (e.g. removing distance-dependent effects from expression data, calculating differential stability estimates; *Hawrylycz et al., 2015*), and for accessing data from the companion Allen Mouse Brain Atlas (e.g. providing interfaces for querying the Allen Mouse API; https://mouse.brain-map.org/; *Lein et al., 2007*).

Although these workflows support the entire range of processing options that we assessed in the current manuscript (*Figure 4c*), we have set the default options for all steps based on best practice recommendations developed in *Arnatkeviciute et al., 2019* and further informed by the results

---

volumetric or surface atlas
individualized or group-level atlas

updated probe-to-gene annotations

intensity-based filtering threshold

probe selection method
donor-specific probe selection

use non-linear MNI coordinates

sample-to-region matching tolerance

sample normalization method

gene normalization method
sample-to-region combination metric
sample-to-region combination method

Regional microarray expression data were obtained from 6 post-mortem brains (1 female, ages 24.0–57.0, 42.50 +/- 13.38) provided by the Allen Human Brain Atlas (AHBA, https://human.brain-map.org; [H2012N]). Data were processed with the abagen toolbox (version X.Y; https://github.com/rmarkello/abagen) using a 83-region volumetric atlas in MNI space.

First, microarray probes were reannotated using data provided by [A2019N]; probes not matched to a valid Entrez ID were discarded. Next, probes were filtered based on their expression intensity relative to background noise [Q2002N], such that probes with intensity less than the background in >=50.00% of samples across donors were discarded. When multiple probes indexed the expression of the same gene, we selected and used the probe with the most consistent pattern of regional variation across donors (i.e., differential stability; [H2015N]), where regions correspond to the structural designations provided in the ontology from the AHBA.

The MNI coordinates of tissue samples were updated to those generated via non-linear registration using the Advanced Normalization Tools (ANTs; https://github.com/chrisfilo/alleninf). Samples were assigned to brain regions in the provided atlas if their MNI coordinates were within 2 mm of a given parcel. To reduce the potential for misassignment, sample-to-region matching was constrained by hemisphere and gross structural divisions (i.e., cortex, subcortex/brainstem, and cerebellum, such that e.g., a sample in the left cortex could only be assigned to an atlas parcel in the left cortex; [A2019N]). All tissue samples not assigned to a brain region in the provided atlas were discarded.

Inter-subject variation was addressed by normalizing tissue sample expression values across genes using a robust sigmoid function [F2013J]. Normalized expression values were then rescaled to the unit interval.

Gene expression values were then normalized across tissue samples using an identical procedure. Samples assigned to the same brain region were averaged separately for each donor and then across donors.

**Figure 5.** Annotated example abagen report. Example of an automatically generated methods section report from the abagen toolbox. Processing steps are shown on the left and the relevant methods text—which is updated when these steps are modified—is shown in the same font color on the right. Reports also include a formatted reference section and relevant equations; these are not shown here for conciseness. Note that some processing steps (e.g. normalizing within structures, missing data handling) are omitted here because they are not run by default (see *Supplementary file 1*).

presented above (see *Supplementary file 1* for a full list). We believe the default settings in abagen will provide a reasonable starting point for researchers beginning to work with the AHBA; however, as we have continually noted, the appropriate choices for some parameters will vary based on research question. As such, to make it easier for researchers to report exactly what parameters they use, we have integrated an automated reporting mechanism into the abagen workflows (*Figure 5*). The generated reports provide manuscript-ready step-by-step documentation describing all the processing done to the AHBA data in the workflow, and are licensed CC0 (https://creativecommons.org/share-your-work/public-domain/cc0/) so that they can be freely used without restriction.

Creation of the toolbox has followed best-practices in software development, including version control, continuous integration testing, and modular code design. To encourage further use by new research groups we provide comprehensive documentation on installing and working with the abagen toolbox online (https://abagen.readthedocs.io/).

## Discussion

In the present report, we introduced the abagen toolbox, an open-source Python library for processing transcriptomic data. Using abagen, we conducted a comprehensive analysis examining whether and how different processing options modify statistical estimates derived from analyses using the AHBA. We investigated how processing pipelines used in the literature compare to those we tested, and provide recommendations for improving standardization and reporting of analyses using the AHBA, highlighting how the abagen toolbox can facilitate future developments in this space.

Testing nearly 750,000 unique processing pipelines, we find that choice of processing parameters can strongly influence statistical estimates derived from analyses of the AHBA, and that these choices interact with the type of analysis performed (*Figure 1*). We observe significant variability with regard to which parameters are most influential, finding that procedures modifying gene expression normalization have a far greater impact on downstream analyses than other processing steps (*Figure 2*). Looking to the literature, we reproduce nine pipelines from published articles and find that, despite notable inconsistencies in their processing choices, there is moderate consistency in their produced statistical estimates (*Figure 3*). We demonstrate, however, that these summary estimates may obscure meaningful differences in gene expression values derived by the pipelines, cautioning researchers to be aware of how analytic choices may impact their findings.

Altogether, the present report provides a comprehensive assessment of how processing variability can impact analyses in the field of imaging transcriptomics. Our results demonstrate how researcher choices (or 'researcher degrees of freedom'; *Simmons et al., 2011*) can play a meaningful role in analyses of the AHBA. However, these findings are not necessarily limited to the AHBA. Indeed, increasing reliance on open-access datasets has begun to reveal unique challenges associated with data reuse (*Thompson et al., 2020*). Improved standardization and reporting among research groups using (and re-using) openly available datasets may help to mitigate some of these challenges. We believe that functionality in the abagen toolbox can support future researchers in overcoming these pitfalls and improve reproducibility in processing and analyzing AHBA data.

Our results also show that not all processing choices are equal: that is, we find a hierarchy of processing parameters, wherein procedures modifying gene normalization have the greatest impact on analyses, followed by steps more broadly influencing the matching of tissue samples to brain regions and finally by parameters that determine probe selection. Furthermore, we find that within processing steps certain parameter choices may lead to more reasonable statistical estimates. In particular, applying some form of gene normalization tends to improve the behavior of processed expression data when compared to instances in which no normalization is applied (*Figure 1*), but there appear to be limited differences in the type of normalization used. Although we only considered cortical tissue samples in the current analyses, we expect that including non-cortical samples would further reinforce these results (*Arnatkeviciute et al., 2019*) known differences in microarray expression values between cortex and subcortical structures will likely emphasize the impact of different normalization procedures across pipelines. Critically, these findings largely agree with previous recommendations developed by *Arnatkeviciute et al., 2019*, and we have chosen default parameter choices for abagen workflows accordingly.

Note that there are some processing steps that should be performed in a specific sequence, and others whose order could potentially be interchanged. For example, intensity-based filtering of probes

must always be performed before probe selection—reversing the order of these operations would, in the majority of cases, be problematic because it would potentially result in the selection of noisy probes to be carried through to analysis. However, the order of other steps (i.e. sample versus gene normalization) could arguably be reversed with no ostensible detriment. This procedural ambiguity is a salient example of the need to standardize workflows.

More broadly, this work builds on increasing efforts to examine the importance of methodological choices and analytical flexibility in human neuroimaging research (*Bhagwat et al., 2021*; *Kharabian Masouleh et al., 2020*; *Oldham et al., 2020*; *Maier-Hein et al., 2017*; *Schilling et al., 2019*; *Carp, 2012*; *Botvinik-Nezer et al., 2020*; *Parkes et al., 2018*; *Ciric et al., 2017*). Thankfully, emerging technical solutions have begun to tackle these issues via the development of tools that aim to abstract away sources of variation (e.g. fMRIPrep, *Esteban et al., 2019*; QSIPrep, *Cieslak et al., 2020*). While results from the present study reinforce the importance of methodological choices in research, abagen draws significant inspiration from these software packages in providing a set of tools designed to overcome such concerns when working with the AHBA.

While the AHBA dataset remains the only one of its kind, the abagen toolbox is designed to be used more broadly as similar datasets become available. That is, the preprocessing functions in abagen can be applied to other microarray expression datasets assuming, for example, availability of stereotactic coordinates. As new imaging transcriptomic datasets are developed and become more widely used, abagen functionality for creating standardized processing pipelines will only become more important. By developing the toolbox openly on GitHub (https://github.com/rmarkello/abagen), it is our hope that abagen can serve as a foundational, community tool for use in imaging transcriptomics research.

One consideration for future work on this topic is that the pipelines tested cover only a portion of the potential variability possible when processing AHBA data (*Table 1*). For example, a growing body of research has begun to examine how choice of brain parcellation may impact imaging analyses (e.g. *Craddock et al., 2012*; *Thirion et al., 2014*; *Messé, 2020*; *Markello and Misic, 2021*). While we only assessed processing pipelines using the Desikan-Killiany atlas, many other atlases have been used with the AHBA and it remains unclear how this variation may impact research findings. We also did not investigate whether donor-specific parcellations may impact analyses, a processing choice used in several published research findings (*Anderson et al., 2020b*; *Romero-Garcia et al., 2018*; *Burt et al., 2018*). Although there is significant evidence suggesting inter-individual variability in brain region definition (e.g. *Gordon et al., 2017*; *Kong et al., 2019*; *Dickie et al., 2018*), the process of generating individualized brain parcellations is fraught with methodological choices and requires careful data processing. Given the quality of the MRI data provided alongside the transcriptomic data in the AHBA—including important differences in scanning protocol and procedures between donors—creating donor-specific parcellations may be a large source of variability between pipelines.

Another limitation of the presented results is that we are unable to make categorical statements about which processing options are 'best' for the AHBA. First, there is no ground truth against which one can assess what the optimal set of processing parameters. One potential solution to this could be to examine the robustness of pipelines based on a leave-one-donor-out strategy (e.g. *Arnatkeviciute et al., 2019*; *Vogel et al., 2020*), wherein analyses are repeated six times, omitting one donor each time, to ensure that none of the donors are unduly influencing analytic estimates. This approach is likely to become more useful as data from more individuals becomes available, but at present may be a worthwhile approach for assessing whether chosen processing parameters are appropriate. More-over, the optimal set of processing parameters may vary based on research question. For instance, in most applications gene normalization is appropriate, as it ensures that downstream analyses are not driven by a small subset of highly expressed genes. However, in other applications it may be desirable to retain the variance contributed by genes to accurately reflect their relative expression levels. For example, many genes in AHBA are not brain-specific, so normalization will amplify their expression patterns, potentially obscuring more relevant expression information. This can be avoided by sub-selecting genes in a hypothesis-driven manner and skipping the normalization step altogether.

Nonetheless, we offer two alternative solutions for researchers who want to continue using the AHBA data. First, similar to the current report, researchers can conduct a comprehensive analysis with the AHBA, running multiple processing pipelines and showing the entire distribution of generated statistical estimates; however, this process can be computationally prohibitive and may impair researchers' abilities to interpret their findings (*Steegen et al., 2016*). A less costly alternative, then,

is for the imaging transcriptomic research community to converge on a set of data-driven processing pipeline for the AHBA that can be used across research groups. We believe the abagen toolbox—with its comprehensive workflows, well-informed default parameter choices, and detailed documentation—can facilitate this process. While we acknowledge that some research groups may have strong reasons for wanting to use specific (i.e. non-default) processing choices, in these instances we urge clear and detailed reporting of the methods used—such as via the automated reporting functionality from the abagen toolbox.

Altogether, the current report highlights the problem of processing variability in analyses using the AHBA, impacting many research studies in the burgeoning field of imaging transcriptomics. We demonstrate how different processing options can influence statistical estimates of analyses relating data from the AHBA to imaging-derived phenotypes, and present the abagen toolbox as a promising potential solution to this issue.

# Materials and methods

## Code and data availability

All code used for data processing, analysis, and figure generation is available on GitHub (https://github.com/netneurolab/markello_transcriptome; *Markello, 2021a* copy archived at swh:1:rev:3abb-c85596a5baacd93e5e9e56c906c9dbb080f3)and directly relies on the following open-source Python packages: IPython (*Perez and Granger, 2007*), Jupyter (*Kluyver et al., 2016*), Matplotlib (*Hunter, 2007*), NiBabel (*Brett et al., 2019*), NumPy (*Oliphant, 2006*; *van der Walt et al., 2011*; *Harris et al., 2020*), Pandas (*McKinney, 2010*), PySurfer (*Waskom et al., 2020*), Scikit-learn (*Pedregosa et al., 2011*), SciPy (*Virtanen et al., 2020*), and Seaborn (*Waskom et al., 2018*).

## Data

### Allen human brain atlas

The Allen Human Brain Atlas (AHBA) is an open-access online resource containing whole-brain microarray gene expression data obtained from post-mortem tissue samples of six adult human donors (https://human.brain-map.org; *Allen Institute for Brain Science, 2013*; *Hawrylycz et al., 2012*). Expression data for over 20,000 genes were sampled from 3702 distinct tissue samples across the six donors (one female, ages 24–57), providing the most spatially comprehensive assay of gene expression in the human brain. Normalized microarray expression data were downloaded for all six donors; RNAseq data were downloaded for the two donors with relevant data.

### Human connectome project

Group-averaged T1w/T2w (a proxy for intracortical myelin) data were downloaded from the S1200 release of the Human Connectome Project (HCP; *Van Essen et al., 2013*) and used without further processing.

### Brain parcellations

All analyses were performed with the Desikan-Killiany atlas (DK; 68 cortical nodes), an anatomical parcellation generated by delineating regions based on gyral boundaries (*Desikan et al., 2006*). To explore the impact of volumetric- versus surface-based parcellations we used a version of the DK atlas in (1) volumetric MNI152, and (2) surface fsaverage5 space; both versions are provided directly with the abagen toolbox. To facilitate comparison between volumetric- and surface-based parcellations, samples from the cerebellum, subcortex and brainstem were omitted.

### The abagen toolbox

Source code for abagen is available on GitHub (https://github.com/rmarkello/abagen) and is provided under the three-clause BSD license (https://opensource.org/licenses/BSD-3-Clause). We have integrated abagen with Zenodo, which generates unique digital object identifiers (DOIs) for each new release of the toolbox (e.g. https://doi.org/10.5281/zenodo.3451463). Researchers can install abagen as a Python package via the PyPi repository (https://pypi.org/project/abagen/), and can access comprehensive online documentation via ReadTheDocs (https://abagen.readthedocs.io/).

## Gene expression pipelines

Most neuroimaging analyses using the AHBA must first convert the 'raw' data into a pre-processed brain region-by-gene expression matrix. To investigate the extent to which different processing procedures might impact downstream analyses, we used abagen to modify 17 distinct processing steps in the generation of region-by-gene matrices from the original AHBA data. Each unique set of these 17 processing choices and parameters constitutes a pipeline, yielding 746,496 unique pipelines. Here, we describe in detail the 17 processing steps and respective methods for each option that we examined in our analyses (refer to *Table 1* for a summary overview of these choices or refer to the abagen documentation for implementation details; https://abagen.readthedocs.io).

## Volumetric or surface atlas

Aggregation of tissue samples from the AHBA into discrete brain regions requires researchers to supply an atlas (or parcellation). There are many brain atlases available for use; however, they typically exist in one of two forms: defined (1) in 3D 'volumetric' space, or (2) in 'surface' space on a 2D representation of the cortical sheet. Many atlases can exist in both of these formats and so beyond the choice of parcellation, researchers must select which representation to use when processing AHBA samples. Choice of atlas may impact how many and which samples are matched to brain regions. In the current manuscript, we examined a volume- and surface-based representation of the Desikan-Killiany atlas (see *Materials and methods: Data*; *Desikan et al., 2006*). Note that both versions of the atlas used in the reported analyses are included with the abagen software distribution.

## Individualized or group-level atlas

There is growing recognition that brain parcellations derived at the group level tend to obscure individual differences in anatomy or function (e.g. *Gordon et al., 2017*; *Kong et al., 2019*; *Dickie et al., 2018*). Researchers working with the AHBA have thus begun to generate donor-specific parcellations, using individualized atlases to match tissue samples to brain regions. The individualization process can vary dramatically depending on whether researchers are using volumetric or surface atlases and whether they are operating in 'native' or standard (i.e. group) space. Because of the immense variability inherent to the individualization process itself, we opted not to explore this parameter in the current manuscript.

## Use non-linear MNI coordinates

With its initial release the AHBA provided stereotactic coordinates for each tissue sample in MNI space (*Fonov et al., 2009*; *Fonov et al., 2011*; *Collins et al., 1999*); however, two of the six donor brains were scanned *in cranio* and coordinates were derived using affine registrations to the MNI template, while the remaining four were scanned ex vivo and a non-linear registration was used to generate coordinates. More recently, *Gorgolewski et al., 2014* used ANTS (*Avants et al., 2011*) to perform a standardized, manually corrected non-linear diffeomorphic registration of all the donor brains to MNI space. Analyses collating tissue samples into distinct brain regions often rely on MNI coordinates to match samples to regions, and researchers must choose whether to use the original coordinates provided with the AHBA or the newer, non-linearly generated coordinates. In the current manuscript, we assessed the impact of using (1) the original MNI coordinates and (2) the updated coordinates from *Gorgolewski et al., 2014*.

## Mirror samples across left-right hemisphere

Only the first two donors included in the AHBA had tissue samples taken from the right hemisphere. Preliminary analyses of these data revealed minimal lateralization of microarray expression, and so samples were collected exclusively from the left hemisphere for the following four donors (*Hawrylycz et al., 2012*; *Hawrylycz et al., 2015*). This irregular sampling resulted in limited spatial coverage of expression in the right hemisphere; to resolve this, some researchers have opted to mirror existing tissue samples across the left-right hemisphere boundary (*Romero-Garcia et al., 2018*). Researchers must decide whether to perform sample mirroring, and, if so, whether they should mirror unilaterally (i.e. only right-to-left *or* left-to-right) or bilaterally (i.e. both right-to-left and left-to-right). In the current manuscript, we assessed (1) no mirroring, (2) left-to-right mirroring, and (3) bilateral mirroring. The

option for mirroring right-to-left was omitted as this is only useful when analyses selectively consider the left hemisphere, not the whole brain.

### Update probe-to-gene annotations

The 60-base-pair probes used to assess microarray expression in the AHBA were annotated with their corresponding gene (or lack thereof) when the data were publicly released. However, as the human reference genome is updated these annotations become increasingly out-of-date. Thus, when researchers choose to use the AHBA data they must decide whether to use the original gene annotations or more recently-generated annotations. In the current manuscript, we assessed using both the original annotations and those generated by *Arnatkeviciute et al., 2019*.

### Intensity-based filtering threshold

Data from the AHBA are provided with information indicating whether the expression of each microarray probe exceeds the expression levels of background signal. Using this information, researchers can choose to perform an intensity-based filtering procedure wherein probes are only considered if their expression levels are greater than background across a specified percentage of tissue samples. In the current manuscript, we considered three degrees of intensity-based filtering: (1) no filtering (all probes used), (2) 25 % filtering (probes used if they exceeded background for more than 25 % of all samples), and (3) median filtering (probes used if they exceeded background for more than 50 % of all samples).

### Inter-areal similarity threshold

The expression value of some tissue samples in the AHBA differ markedly from all other samples in the dataset. While this could be driven by real spatial variability in expression values throughout the brain, it is also possible that this variability is artifactual. Researchers can opt to assess the inter-areal similarity of tissue samples, quantifying those that differ from the rest by a given threshold, and remove them from consideration. To our knowledge, this processing step has only been implemented in a single research study (*Burt et al., 2018*), and as such we do not consider it in the current manuscript.

### Probe selection method

The probes used to measure microarray expression levels in the AHBA are often redundant; that is, there are frequently several probes indexing the same gene. Thus, at some point researchers must transition from measuring *probe* expression levels to measuring *gene* expression levels. Effectively, this means selecting from or condensing the redundant probes for each gene. There have been at least eight methods proposed in the literature for this process, including selecting a single probe with the (1) max intensity across samples, (2) max variance across samples, (3) highest loading on the first principal components across samples, (4) highest correlation to other probes (or max intensity across samples when only two probes exist), (5) highest correlation to other probes (or max variance across samples when only two probes exist), (6) highest differential stability across donors, (7) highest fidelity to simultaneously-acquired RNAseq data, or (8) simply averaging all probes indexing the same gene. In the current manuscript we only consider six of the most commonly-applied methods (i.e. 1, 4, 5, 6, 7, and 8); the other methods (i.e. 2 and 3) have only been reported in a single research study (*Negi and Guda, 2017* and *Parkes et al., 2017*, respectively) and as such we do not consider them.

### Donor-specific probe selection

Probe selection (described above) often requires applying some selection criterion to gene expression levels across tissue samples. For these methods, the specified criterion can be measured across donors (i.e. aggregating tissues samples from donors) or independently for each donor. The latter case—performing probe selection independently for each donor—allows for two additional options: (1) using whichever probe is chosen for each donor, even if it differs from the other donors, or (2) using the most-commonly selected probe for all donors. In the current manuscript, we considered all three of these options: (1) aggregating samples across donors, (2) performing probe selection independently for each donor, and (3) using the most commonly-selected probe across donors.

## Missing data method

Due to the irregular spatial sampling of data in the AHBA some brain regions may not be assigned any corresponding microarray expression data. Researchers can opt to simply omit these regions from subsequent analyses; however, in some cases, this is not desirable as the spatial distribution of the missing samples may not be random and discarding them may bias resulting estimates. Two options for handling missing data have been proposed in the literature, including filling missing regions with expression data from nearby regions (i.e. nearest-neighbors interpolation; *Whitaker et al., 2016*), or interpolating data in missing regions based on nearby samples (i.e. linear interpolation; *Burt et al., 2018*). In the current manuscript, we tested two options: (1) omit brain regions with missing data entirely from subsequent analyses, and (2) fill missing data with expression values using nearest-neighbors interpolation. Linear interpolation has been sparingly used in the published literature (e.g. *Burt et al., 2018*; *Romero-Garcia et al., 2018*) and carries an increase in computational cost (approximately an order of magnitude higher than nearest neighbors interpolation); as such, we do not consider it in the current manuscript.

## Sample-to-region matching tolerance

### Volumetric atlases

While most tissue samples from the AHBA will fall directly within the brain regions delineated by most parcellations, some samples may fall outside the boundaries of these regions. Researchers can nonetheless choose to permit assigning these nearby samples to a given region, but will often set a distance threshold beyond which samples cannot be assigned. In the current manuscript, we considered three distance tolerances: 0 mm (i.e. samples must fall exactly within a region), 1 mm, and 2 mm.

### Surface atlases

Because tissue samples from the AHBA are defined in volumetric space, matching them to parcels defined on a surface-based atlas requires different considerations than with volumetric atlases. Notably, all samples will have non-zero distances from surface vertices; therefore, when matching to surface atlases distance thresholds are generally considered in terms of standard deviations (*Burt et al., 2018*; *Anderson et al., 2020b*). In this way, all samples are matched to the surface and then those that are more than the specified standard deviation(s) above the mean away from the surface are excluded. In the current manuscript we tested three standard deviation distance tolerances: 0 s.d. (i.e. all samples farther than the average distance are excluded), 1 s.d., and 2 s.d.

## Sample normalization method

Prior to aggregating microarray expression data across donors, researchers can optionally normalize the microarray expression data for each tissue sample across all represented genes (i.e., perform row-wise normalization). This procedure can account for between-sample differences in gene expression potentially driven by measurement errors. There is a number of techniques that have been proposed to normalize expression values; however, in the current manuscript, we considered three normalization methods: (1) no normalization, (2) a z-score transform, and (3) a scaled robust sigmoid transform (*Fulcher et al., 2013*).

## Gene normalization method

Prior to aggregating microarray expression data across donors, researchers can optionally normalize the microarray expression data for each represented gene across tissue samples (i.e. perform column-wise normalization). This procedure can account for inter-individual (donor-specific) differences in gene expression data, which remain present in the AHBA despite batch corrections performed by the Allen Institute prior to releasing the data. In the current manuscript, we considered three normalization methods: (1) no normalization, (2) a z-score transform, and (3) a scaled robust sigmoid transform (*Fulcher et al., 2013*).

## Normalizing only matched samples

Due to choices in other processing steps (e.g. *Volume- or surface-based atlas*, *Sample-to-region matching tolerance*) some tissue samples from the AHBA may not be assigned to any region in a given brain atlas. During gene normalization, where expression from each gene is normalized across

tissue samples, researchers must decide whether to use (1) only those tissue samples matched to brain regions, or (2) the entire corpus of tissue samples, irrespective of whether they will be included in the final, processed regional expression matrix. In the current manuscript we consider both of these options.

### Normalizing discrete structures
There is known variation in gene expression values between tissue samples taken from distinct structural classes (i.e. samples taken from neocortex may have different expression values than those from the brainstem). When performing gene normalization researchers can opt to normalize (1) across all samples irrespective of the structure from which they derive or (2) independently for samples taken from different brain structures. Although the brain atlas used in the current manuscript represents only cortical parcels, this processing choice can interact with *Normalizing only matched samples* to impact resulting expression values and we therefore test both options.

Note that in the abagen toolbox structural classes are operationalized as: (1) cortex, (2) subcortex and brainstem, (3) cerebellum, and (4) white matter. Subcortex and brainstem are considered as one class because neuroanatomical delineation between these regions are widely contested and expression values in these regions tend to be more similar to one another than to other regions (i.e. data-driven clustering of samples tends to assign subcortical and brainstem samples together).

### Sample-to-region combination method
Once tissue samples have been assigned to brain regions they need to be combined to generate a single expression profile; however, due to sampling differences between donors, some donors may have more tissue samples assigned to a given brain region than others. Thus, researchers must decide whether to aggregate samples (1) within each brain region independently for each donor and then across donors, or (2) simultaneously across all donors. In the latter case, donors with a higher number of samples matched to a region will contribute more to the expression profile of a given region (*Arnatkeviciute et al., 2019*). In the current manuscript, we test both of these options.

### Sample-to-region combination metric
When aggregating tissue samples into brain regions researchers must decide what aggregation metric they want to use. Although any statistical estimate could be considered, in practice an estimate of central tendency such as the mean expression values across tissue samples is most applicable. In the current manuscript, we test aggregation with both the (1) mean and (2) median.

### Analytic approaches
Prototypical analyses relying on parcellated microarray expression data from the AHBA fall into three broad categories (*Fornito et al., 2019*):

1. *Correlated gene expression*: Examining the correlation between distinct brain regions across genes (i.e. using the region-by-region correlation matrix);
2. *Gene co-expression*: Examining the correlation between gene expression profiles across brain regions (i.e. using the gene-by-gene correlation matrix); or,
3. *Regional gene expression*: Examining the expression profile of one (or more) genes across brain regions (i.e. using selected columns of the region-by-gene expression matrix).

In order to examine the interaction between processing options and analytic method, we performed one analysis from each of these three categories, described below, for every output of the 746,496 processing pipelines.

### Correlated gene expression
Researchers have reliably found a relationship between correlated gene expression in the brain and the distance between brain regions: that is, brain regions that are farther away from one another tend to have less similar gene expression profiles (*Richiardi et al., 2015*; *Richiardi et al., 2017*; *Krienen et al., 2016*; *Vértes et al., 2016*; *Arnatkeviciute et al., 2019*). In order to examine the impact of processing choices on this relationship, we computed the Spearman correlation between the upper triangle of the regional distance matrix (Euclidean distance between brain regions) and the upper triangle of each correlated gene expression matrix (*Figure 1a*, left). Brain regions for which no gene

expression data were available (dependent on pipeline options) were not included in the correlation. Note that this relationship is likely exponential (*Arnatkeviciute et al., 2019*); however, we calculated the Spearman coefficient as it is more computationally tractable and it should exhibit similar variability across pipelines.

## Gene co-expression

Researchers have previously shown that gene expression in the brain tends to organize into functionally defined communities or modules (*Oldham et al., 2008*; *Hawrylycz et al., 2012*). We examined the extent to which functional gene modules derived from a separate transcriptomic dataset (*Oldham et al., 2008*) mapped onto the gene co-expression matrices generated from the different processing pipelines. For each gene-by-gene matrix, we calculated the silhouette score (*Rousseeuw, 1987*) of the gene modules on a modified version of gene co-expression matrix (calculating Euclidean distance between genes instead of gene correlations; *Figure 1a*, middle) via:

$$s = \frac{1}{N} \sum_{i=1}^{N} \frac{b(i) - a(i)}{\max\{a(i), b(i)\}}$$

where $a(i)$ is the average distance of a data point  to all other data points in the same cluster, $b(i)$ is the mean distance of data point  to the nearest neighboring cluster, and N is the total number of data points. The final silhouette score $s$ ranges from –1 to +1, where positive values indicate assortative and negative values indicate disassortative clusters.

Note that the original gene modules were defined using a weighted gene co-expression network analysis (WGCNA), which generally requires performing additional processing steps on the gene co-expression matrix. Since we used the raw gene co-expression matrix in the current analysis, we expect lower silhouette scores than those reported in the initial manuscript where the gene communities were initially defined; however, the *variance* in scores between pipelines should not be significantly impacted by this choice.

## Regional gene expression

Researchers recently highlighted how the principal component of gene expression in the brain closely mirrors the spatial variation observed in MRI-derived T1w/T2w measurements (typically used as a proxy for myelination; *Burt et al., 2018*). We examined whether this relationship was present across the outputs of the different pipelines, measuring the Spearman correlation between the T1w/T2w ratio and the first principal component of the regional gene expression matrix (*Figure 1a*, right). Regional gene expression matrices were mean-centered prior to extraction of the principal component.

## Assessing pipeline impact

In order to examine the impact of each processing option on the resulting analyses, we calculated a difference score, measuring the extent to which changing each option—holding all other options constant—influenced the derived metrics (i.e. correlation, silhouette score). When there were only two choices for a given option the impact was calculated as the absolute value of the difference between the two choices. When there were more than two choices and choices were ordinal (e.g. sample-to-region matching tolerance) the impact was calculated as the average of the absolute value of the difference between adjacent choices. When there were more than two choices and the choices were categorical (e.g. probe selection method) the impact was calculated as the average of the absolute value of the difference between all combinations of choices. These calculations yielded a distribution of 'impact' estimates (i.e. change scores) for each processing option; we represented the final impact score for each processing option as the average of these distributions, taken independently for each of the three analyses. Impact estimates were rank-ordered (where the most impactful parameter was given a rank of one, the second most impactful a rank of two, and so on) to enable direct comparison across the different statistical estimates derived from the three analyses.

## Pipeline dimensionality reduction

To investigate qualitative differences between the processing pipelines we performed a principal components analysis (PCA) on the matrix of estimates from the three statistical analyses (i.e. the 746,496 × 3 matrix). We mean-centered the columns of the matrix and extracted the first two principal

components, examining how pipeline scores were distributed along these two components in relation to different processing options. These principal component highlight the closeness of the estimate generated by each pipeline along the dimensions of maximum statistical variation; that is, two pipelines that are closer together in the reduced-dimension space yielded more similar statistical estimates than two pipelines that are farther apart.

## Reproducing pipelines from the literature

Although all the processing options explored in the current manuscript are reasonable or viable choices that researchers could make when preparing the AHBA for analysis, in reality these have not all been used in the published literature. In order to examine how pipelines used in the literature compared to those that we assessed, we selected nine articles that relied on data from the AHBA to support a primary research finding and reproduced their processing pipelines in abagen (*Hawrylycz et al., 2015*; *French and Paus, 2015*; *Whitaker et al., 2016*; *Krienen et al., 2016*; *Anderson et al., 2018*; *Burt et al., 2018*; *Romero-Garcia et al., 2018*; *Anderson et al., 2020b*; *Liu et al., 2020*). Note that these articles used a variety of parcellations and so to ensure comparability across pipelines we standardized this parameter, using the Desikan-Killiany atlas in all instances. One parameter that we did not assess in the pipelines explored in the current manuscript—whether to use individualized, donor-specific parcellations or a group-level atlas—was frequently varied in the published pipelines. Thus, when reproducing pipelines that called for individualized volumetric atlases we relied on the donor-specific Desikan-Killiany parcellations provided by *Arnatkeviciute et al., 2019*; when reproducing pipelines with individualized surface atlases we relied on the donor-specific Desikan-Killiany parcellations provided by *Romero-Garcia et al., 2018*.

As not all of the original manuscripts detailed the processing choices for each of the 17 steps in the abagen workflow, when specific parameter choices were omitted we either: (1) used the default setting if the parameter was required (e.g. using the mean for the 'sample-to-region combination metric', since all pipelines must combine samples to regions), or (2) omitted the processing step entirely if it is an optional step (e.g. not performing any gene normalization).

## Acknowledgements

We thank Vincent Bazinet, Elizabeth DuPre, Justine Hansen, Golia Shafiei, Laura Suárez, and Bertha Vázquez-Rodríguez for their comments and suggestions. This research was undertaken thanks in part to funding from the Canada First Research Excellence Fund, awarded to McGill University for the Healthy Brains for Healthy Lives initiative. This work was supported in part by funding provided by Brain Canada, in partnership with Health Canada, for the Canadian Open Neuroscience Platform initiative. RDM acknowledges support from the Fonds du Recherche Québec - Nature et Technologies and the Canadian Open Neuroscience Platform. BM acknowledges support from the Natural Sciences and Engineering Research Council of Canada (NSERC Discovery Grant RGPIN #017–04265) and from the Canada Research Chairs Program. AF was supported by the Sylvia and Charles Viertel Foundation and National Health and Medical Research Council (ID: 3274306). J-BP was partially funded by National Institutes of Health (NIH) NIH-NIBIB P41 EB019936 (ReproNim) NIH-NIMH R01 MH083320 (CANDIShare) and NIH RF1 MH120021 (NIDM), the National Institute Of Mental Health of the NIH under Award Number R01MH096906 (Neurosynth), and by Natural Sciences and Engineering Research Council of Canada (NSERC).

## Additional information

### Funding

| Funder | Grant reference number | Author |
|---|---|---|
| Natural Sciences and Engineering Research Council of Canada | 017-04265 | Bratislav Misic |

| Funder | Grant reference number | Author |
| --- | --- | --- |
| National Health and Medical Research Council | 3274306 | Alex Fornito |
| National Institutes of Health | NIH-NIBIB P41 EB019936 | Jean-Baptiste Poline |

The funders had no role in study design, data collection and interpretation, or the decision to submit the work for publication.

## Author contributions

Ross D Markello, Conceptualization, Formal analysis, Investigation, Methodology, Resources, Software, Validation, Visualization, Writing – original draft, Writing – review and editing; Aurina Arnatkeviciute, Conceptualization, Data curation, Software, Writing – review and editing; Jean-Baptiste Poline, Alex Fornito, Conceptualization, Writing – review and editing; Ben D Fulcher, Conceptualization, Software, Writing – review and editing; Bratislav Misic, Conceptualization, Project administration, Supervision, Visualization, Writing – original draft, Writing – review and editing

## Author ORCIDs

Ross D Markello http://orcid.org/0000-0003-1057-1336
Jean-Baptiste Poline http://orcid.org/0000-0002-9794-749X
Ben D Fulcher http://orcid.org/0000-0002-3003-4055
Bratislav Misic http://orcid.org/0000-0003-0307-2862

## Decision letter and Author response

Decision letter https://doi.org/10.7554/eLife.72129.sa1
Author response https://doi.org/10.7554/eLife.72129.sa2

# Additional files

## Supplementary files

• Transparent reporting form

• Supplementary file 1. Default abagen pipeline options. The default settings for the 17 processing steps considered when processing the AHBA data with abagen. An entry of '—' indicates that this is a required, user-supplied parameter. A blank entry indicates that the processing step is not implemented by default. Refer to *Table 1* and Methods: Gene expression pipelines for further details.

## Data availability

All datasets used in this study are publicly available. Detailed information about the datasets and how to access them are described in the manuscript.

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
