## [Editor Report]

This paper will be of interest to scientists studying the large-scale transcriptomic organization of the human brain, and in particular those who have used or plan to use the Allen Human Brain Atlas dataset. The study is well-motivated and novel. The most striking finding is the magnitude of variability that is introduced by different data processing decisions. The open-source software described in this study is comprehensive, well documented, and is an important contribution to the field.

---

## [Decision Letter]

**Decision letter after peer review:**

Thank you for submitting your article "Standardizing workflows in imaging transcriptomics with the abagen toolbox" for consideration by *eLife*. Your article has been reviewed by 3 peer reviewers, including Saad Jbabdi as Reviewing Editor and Reviewer #1, and the evaluation has been overseen by Tamar Makin as the Senior Editor. The following individuals involved in review of your submission have agreed to reveal their identity: Joshua Burt (Reviewer #2); Michael J Hawrylycz (Reviewer #3).

Essential revisions:

As you will see, all reviewers are enthusiastic about the paper and the accompanying toolbox. The main things the reviewers are asking for are:

1) Further comments on how the tool can be used to better converge on more strongly interpretable results.

2) More detailed description of the toolbox output.

*Reviewer #1 (Recommendations for the authors):*

This is a first for me but I don't have anything that I'd like the authors to change.

*Reviewer #2 (Recommendations for the authors):*

Are region-region distances geodesic or Euclidean?

P13 – typo: "asses" (you meant assess).

Figure 3 – Panel C might benefit from a schematic. I personally found it somewhat difficult to understand exactly what was being shown.

The number of pipelines tested is written at least two different ways (746,946 and 746,496).

*Reviewer #3 (Recommendations for the authors):*If length becomes an issue, one might reduce some of the description of the many challenges in the field and data analysis which are more well known.

---

## [Author Response]

Reviewer #2 (Recommendations for the authors):Are region-region distances geodesic or Euclidean?

In our analysis of the distance-dependent relationship of gene expression ("correlated gene expression" and "CGE" in the manuscript) we used the Euclidean distance between parcel centroids. For volumetric atlases, parcel centroids were calculated as the center-of-mass of the voxels within each parcel; for surface-based atlases, parcel centroids were calculated as the coordinate of the vertex with the minimum Euclidean distance to the center-of-mass of the vertices within each parcel. We have clarified this in the revised manuscript (“Results” section, “Correlated gene expression” subsection):

“We assessed this relationship by extracting the upper triangle of the correlated gene expression matrices and correlating them with the upper triangle of a regional distance matrix, derived by computing the average Euclidean distance between brain region centroids in the Desikan-Killiany atlas.”

P13 – typo: "asses" (you meant assess).

Fixed!

Figure 3 – Panel C might benefit from a schematic. I personally found it somewhat difficult to understand exactly what was being shown.

We have attempted to clarify Figure 3c by updating the figure labels and caption to be more explicit that the correlations shown are between the maps depicted in Figure 3b.

“(c) The Pearson correlation between the cortical somatostatin (SST) maps generated by the nine pipelines shown in panel (b).”

The number of pipelines tested is written at least two different ways (746,946 and 746,496).

We have updated the number of pipelines throughout the manuscript to the correct value of 746,496.